# Orphan Drugs in Neurology—A Narrative Review

**DOI:** 10.3390/jpm13030420

**Published:** 2023-02-26

**Authors:** Carmen Adella Sirbu, Raluca Ivan, Francois Jerome Authier, Florentina Ionita-Radu, Dragos Catalin Jianu, Octavian Vasiliu, Ciprian Constantin, Sorin Tuță

**Affiliations:** 1Department of Neurology, ‘Dr. Carol Davila’ Central Military Emergency University Hospital, 010242 Bucharest, Romania; 2Centre for Cognitive Research in Neuropsychiatric Pathology (Neuropsy-Cog), Department of Neurology, Faculty of Medicine, “Victor Babeș” University of Medicine and Pharmacy, 300041 Timisoara, Romania; 3Clinical Neurosciences Department, University of Medicine and Pharmacy “Carol Davila” Bucharest, 050474 Bucharest, Romania; 4INSERM U955-Team Relaix, Faculty of Health, Paris Est-Creteil University, 94010 Créteil, France; 5Neuromuscular Reference Center, Henri Mondor University Hospital, APHP, 94000 Créteil, France; 6Department of Gastroenterology, “Carol Davila” Central Military Emergency University Hospital, 010825 Bucharest, Romania; 7Department of Gastroenterology, “Carol Davila” University of Medicine and Pharmacy, 020021 Bucharest, Romania; 8Neurology Department, Victor Babeș University of Medicine and Pharmacy, 300041 Timisoara, Romania; 9Department of Psychiatry, ‘Dr. Carol Davila’ Central Military Emergency University Hospital, 010825 Bucharest, Romania; 10Department of Diabetes, Nutrition, and Metabolic Diseases, Titu Maiorescu University, 031593 Bucharest, Romania; 11Department of Neurology, National Institute of Neurology and Neurovascular Diseases, 041914 Bucharest, Romania

**Keywords:** orphan diseases, spinal muscular atrophy, myasthenia, fabry disease, acute hepatic porphyria, non-dystrophic myotonias, hereditary transthyretin-mediated amyloidosis, duchenne muscular dystrophy, narcolepsy, pompe disease

## Abstract

Background and aims: Orphan diseases, or rare diseases, are defined in Europe as diseases that affect less than 5 out of every 10,000 citizens. Given the small number of cases and the lack of profit potential, pharmaceutical companies have not invested much in the development of possible treatments. However, over the last few years, new therapies for rare diseases have emerged, giving physicians a chance to offer personalized treatment. With this paper, we aim to present some of the orphan neurological diseases for which new drugs have been developed lately. Methods: We have conducted a literature review of the papers concerning rare diseases and their treatment, and we have analyzed the existing studies for each orphan drug. For this purpose, we have used the Google Scholar search engine and the Orphanet. We have selected the studies published in the last 15 years. Results. Since the formation of the National Organization for Rare Diseases, the Orphan Drug Act, and the National Institutes of Health Office of Rare Diseases, pharmacological companies have made a lot of progress concerning the development of new drugs. Therefore, diseases that until recently were without therapeutic solutions benefit today from personalized treatment. We have detailed in our study over 15 neurological and systemic diseases with neurological implications, for which the last 10–15 years have brought important innovations regarding their treatment. Conclusions: Many steps have been taken towards the treatment of these patients, and the humanity and professionalism of the pharmaceutical companies, along with the constant support of the patient’s associations for rare diseases, have led to the discovery of new treatments and useful future findings.

## 1. Introduction

Orphan diseases are rare diseases, defined in Europe as having an incidence lower than 5 per 10,000 citizens. It is not unusual for a family doctor to encounter less than one such case per year. Because of this, there is also a delay in the diagnosis process and the decision-making process, with studies showing that approximately half of the patients with rare diseases receive on average one misdiagnosis [1]. Some of the pathologies may be congenital and present from birth, while others appear during adulthood. There are rare new diseases being described every week in the medical literature, with a total of over 6000 orphan diseases. Because of the lack of profit potential and the costs and logistics necessary to organize clinical trials for these pathologies, the development of specific therapies is not a priority for pharmaceutical companies [2]. Beginning in 1982, signs of progress were made concerning the research in this field, with the foundation of the National Organization for Rare Diseases (NORD) and the approval of the Orphan Drug Act by the Congress of the United States. These encouraged the pharmaceutical companies by providing tax relief to those who participated in the research for new orphan drugs. It would also assure a 7-year exclusivity for the product if used for the treatment of an orphan disease [3].

With this paper, we aim to realize a literature review of some of the orphan drugs used in rare diseases with neurological implications.

## 2. Materials and Methods

We used the Orphan Diseases database in order to identify some of the most important diseases and their orphan drugs. We performed an extensive literature review of the existing studies that analyzed their efficacy and safety, and we cited the ones that brought the most novelties regarding the treatment. We used PubMed and Google Scholar, and we searched for the following terms: rare disease(s), orphan drug(s), orphan disease(s), and neurological disorder(s). We have selected the studies published in the last 15 years. We collected data, such as general information about the disease, diagnosis, clinical features, available treatments, orphan drugs, their mechanisms of action, and adverse reactions, and we showed the progress that the last few years have brought concerning the available resources.

## 3. Results

### 3.1. Main Orphan Neurological Diseases and Their Treatment

#### 3.1.1. Neuromyelitis Optica Spectrum Disorder (NMOSD)

Its prevalence ranges from 0.37 to 10 per 100,000, with a female predominance and a median age of onset of 32 to 41 years. Because of the small number of cases reported, it is considered a rare disease [4]. This is defined as a chronic, autoimmune-mediated disease, that affects the nervous system, damaging mainly the optic nerves and the spinal cord. Usually, the disability results from severe demyelination and extensive axonal damage [5]. The distinctive characteristic of this disease is the presence of antibodies against the aquaporin-4 water channel (AQP4), permitting the clinician to differentiate it from a severe form of multiple sclerosis [6]. The diagnosis includes clinical criteria and MRI findings that show alterations in the optic nerve, the spinal cord, the area postrema, or other sites in the brainstem and diencephalon. If the serological testing is unavailable or the antigens for AQP4 are negative, more extensive clinical and imagistic criteria need to be detected [6]. Cases of negative AQP4-IgG antibodies but with anti-myelin oligodendrocyte glycoprotein (MOG) are also mentioned in the literature [5]. Its pathology also involves interleukin-6, which promotes the formation of proinflammatory type 17 helper T cells and furthers the differentiation of B cells into plasmablasts that produce AQP4-IgG antibodies [7]. Concerning the clinical presentation, the main symptoms of NMOSD are acute attacks of bilateral optic neuritis, with frequent involvement of the optic chiasm, that can eventually lead to vision loss, and, because of the transverse myelitis, a complete or partial spinal cord syndrome, characterized by spasticity, paraparesis or quadriparesis, sensory loss, and often bladder dysfunction. Another common symptom is pain that involves the legs and trunk. Sometimes the patients may also present the Lhermitte sign. As a result of the implication of the aria postrema, patients can present with intractable hiccups, nausea, or vomiting [4,6]. These attacks have different grades of recovery over weeks or months. Other clinical features mentioned in the literature are excessive somnolence or narcolepsy, reversible posterior encephalopathy syndrome, hypothalamic dysfunction, autonomic manifestations (hypotension, bradycardia, hypothermia), and sometimes seizures, especially in children. There are also a few cases in which muscle involvement is also described [5]. The investigations include spinal cord MRI, which shows lesions that extend longitudinally for three or more vertebral segments, best observed on the T2-weighted sequences. These modifications are present in about 60% of the cases, and the lesions can also affect the medulla. There is also gadolinium enhancement, which tends to disappear with treatment. The brain MRI and the MRI of the orbits may be normal or show extensive gadolinium enhancement of the optic nerves. The lesions often involve the central medulla, the hypothalamus, the diencephalon, or the subcortical white matter, causing difficulties in the differential diagnosis of multiple sclerosis [5]. When suspected, the dosage of the AQP4-IgG antibodies is a key feature, with a negative result and high clinical suspicion requiring retesting of these patients. As mentioned before, a few patients that do not present these antibodies may be positive for antibodies against MOG [5]. Lumbar puncture and optical coherence tomography are not required for the establishment of the diagnosis but can be helpful to distinguish NMOSD from other diseases, such as MS [5].

Regarding its treatment, considering its autoimmune origins, the acute attacks are initially managed with high doses of glucocorticoids (1 g daily, for three to five days of methylprednisolone). As a rescue treatment for those who do not respond, plasma exchange can be suggested (1 exchange daily, 7 exchanges total). Concerning the prevention of attacks, not many specific therapies have been developed so far, given the rarity of this disease.

Lately, progress has been made concerning the available specific immunotherapies, with the approval of Eculizumab in 2007 by the European Medicines Agency (EMA) and in 2019 by the US Food and Drug Administration (FDA), and Satralizumab in 2020 by the FDA and in 2021 by the EMA. When not available, clinicians use off-label therapies such as rituximab or tocilizumab [5].

Eculizumab was first approved by the FDA for the reduction of the destruction of red blood cells in adults with paroxysmal nocturnal hemoglobinuria, for children with atypical hemolytic uremic syndrome, and for adults with Myasthenia Gravis who were positive for anti-acetylcholine receptor antibodies [8]. Eculizumab is a humanized antibody that targets the complement C5, inhibiting its cleavage to C5a and C5b and the formation of the C5-induced membrane attack complex (C5b-9). Therefore, it diminishes the inflammation, the subsequent astrocyte necrosis, the death of oligodendrocytes and neurons, and the damage to the blood-brain barrier. It is present as a vial of 30 mL that contains 300 mg of active product (10 mg/mL). In the end, it is diluted, with the final concentration being 5 mg/mL. For the initial phase, 900 mg of Eculizumab is administered slowly (over 25–45 min, intravenously, every week for the first 4 weeks). After this period, in the 5th week, it is administered as intravenous infusions of 1200 mg in 25–45 min and then 1200 mg of intravenous infusion every 14 ± 2 days [9]. The main adverse reactions are described in Table 1. There are two studies concerning the efficacy and safety of this product for its usage in the treatment of NMOSD, one being a phase III, double-blind, randomized, placebo-controlled study (ECU-NMO-301) and another a phase III, open-label, extension trial (ECU-NMO-302) [9].

The ECU-NMO-301 study gathered data from 143 patients who tested positive for AQP4-IgG antibodies, and it compared the efficacy of Eculizumab with a placebo. The patients could also receive background immunosuppressants, except for Rituximab and Mitoxantrone. The patients treated with Eculizumab present a relative reduction of 95.5% in the annualized relapse rate, compared to the placebo. Additionally, those patients presented reduced annualized rates of hospitalization (0.04 for active drug versus 0.31 for placebo), of corticosteroids usage to treat acute relapses (0.07 versus 0.42), and of the necessity of plasma exchange treatment (0.02 versus 0.19) [9]. However, it did not demonstrate a statistically significant reduction in the EDSS score compared to the placebo [10]. The ECU-NMO-302 study analyzed 119 patients and had as its target the evaluation of long-term safety for the patients who completed ECU-NMO-301 and continued to receive Eculizumab [10]. It demonstrated a reduction in the annualized relapse rate after 24 months of the screening conducted in Study ECU-NMO-301. The most common change in the background treatment was the reduction of immunotherapy doses, which occurred in 18.5% of patients, and 6.7% stopped it completely [9].

Satralizumab is a humanized monoclonal antibody that blocks the action of interleukin 6, which has been proven to cause inflammation. To do this, it binds to the membrane-bound and soluble interleukin-6 receptors [7]. It presents as a pre-filled syringe that contains 120 mg of active substance in 1 mL. It may be administered in monotherapy or in combination with immunosuppressive therapy, with the loading dose being 120 mg subcutaneous (sc) injection every two weeks for the first three administrations and then every four weeks for maintenance [11]. The main adverse reactions are described in Table 1.

This drug’s efficacy and profile of safety were evaluated in two pivotal phase III clinical trials (Studies BN40898 and BN40900) in patients diagnosed with NMOSD, both seropositive and seronegative for AQP4-IgG antibodies [11]. The first study (BN40898) included 83 patients aged 12–74 years and evaluated the effect of the drug in combination with other immunosuppressive therapy (azathioprine or mycophenolate) compared to a placebo. In the AQP4-IgG seropositive patients, the drug reduced the relapses by 79% (hazard ratio, HR [95% CI]: 0.21 [0.06–0.75]), and it reduced the need for rescue therapy (corticosteroids or plasma exchange) by 61% (odds ratio [OR] = 0.3930, 95% CI: 0.1343–1.1502; *p* = 0.0883). Additionally, it reduced the risk of increasing the EDSS by two points from the initial assessment by 85% (HR = 0.15, 95% CI: 0.02–1.25; *p* = 0.0441) [11]. The second study (BN40900) included 95 adults, aged 18–74 years old, and evaluated the effect of Satralizumab as monotherapy compared to a placebo. In the AQP4-IgG seropositive patients, the risk of relapse was reduced by 74% (HR [95% CI]: 0.26 [0.11–0.63]). The use of the drug reduced the need for rescue therapy by 74% (OR = 0.2617, 95% CI: 0.0862–0.7943; *p* = 0.0180) and the risk of an increased EDSS by 2 points by 79% (HR = 0.21, 95% CI: 0.05–0.91; *p* = 0.0231) [11]. No notable efficacy was reported for the AQP4-IgG seronegative patients in any study [11].

The evolution of the disease is marked by progressive deterioration, with recurrent visual, motor, sensory, and bladder deficits, and even though progress has been made regarding its treatment, the fact that these drugs are considered orphan drugs and the small number of cases registered so far generate high costs of production and therefore high costs for a specific personalized treatment.

#### 3.1.2. Spinal Muscular Atrophy (SMA) Type 1, Type 2, or Type 3

Spinal muscular atrophy is a rare autosomal recessive neuromuscular disease, with an incidence of 1 in 10,000 cases [12]. It is defined as a degeneration of the alpha motor neurons from the anterior horn cells in the spinal cord or of the motor nuclei present in the brainstem, the result being progressive muscular weakness and atrophy. Based on the age of onset and the clinical manifestations, there are 5 subtypes of the disease, noted from 0 to 4 [13]. It is caused by a genetic defect localized in the survival motor neuron (SMN) gene [12]. The SMN1 gene is presented on chromosome 5q13.2, is responsible for the synthesis of mRNA in the motor neurons and can also inhibit apoptosis. Biallelic deletions or mutations in this gene (the most commonly described is the homozygous deletion of exon 7) cause a deficit of the SMN1 protein. There is a second gene, SMN2, which is 99% identical with the SMN1 gene and located within an inverted duplication on chromosome 5q13.2. The difference between the 2 is a C-to-T transition in exon 7 of SMN2. This leads to the production of a nonfunctional SMN protein. However, 10–15 % of SMN2 can produce functional, full-length SMN protein. Therefore, its synthesis can partially compensate for the loss of the SMN1 gene, and the number of SMN2 copies, which varies from 0 to 8, generates different phenotypes; the higher the number of copies, the milder the form of the disease. Cases of SMA that are not generated by the before-mentioned mechanisms are also described in the literature and have the name of non-5q spinal muscular atrophies [13].

SMA type 0 is described in the extended classification, with signs being present from birth. The mother can describe the decrease in or loss of movement of the fetus in late pregnancy. They usually present with general hypotonia and weakness, areflexia, facial diplegia, congenital heart disease, and arthrogryposis, and may succumb by the age of six months due to respiratory failure [13].

SMA type 1, or infantile SMA, is encountered in infants, with signs that appear after birth and before 6 months of age. The children may appear normal at birth until the onset of syndromes. They present severe hypotonia, usually symmetrical and affecting more the proximal muscles, lack of control for the head muscles, poor spontaneous mobility, and no antigravitational limb movements. The muscles of the face are spared, and they present paradoxical breathing due to the unequal involvement of the diaphragm and the intercostal muscles. They also present abolished or diminished deep tendon reflexes, tongue fasciculation, and swallowing defects. Most of these patients have two or three copies of the SMN2 gene [12].

SMA type 2 has an onset between 7 and 18 months. The children may support their position, but they do not acquire the ability to walk independently. They can cause weakness of the masticatory muscles, but the eye and face muscles are spared. These patients can develop restrictive lung diseases because of the combination of respiratory muscle weakness and scoliosis. They usually present three copies of the SMN2 gene [13].

SMA type 3, or the juvenile form, appears between 18 months and adulthood. Usually, the patients acquire the ability to stand alone, but the debut of the disease is characterized by progressive weakness of the legs more than the arms, with subsequent falls and trouble climbing the stairs. As the illness advances, they can lose their ability to stand, and they will require a wheelchair. They generally present three to four copies of the SMN2 gene [13].

SMA type 4 is the mildest form, with an onset >18 years old. The patients can stand, walk, and do not have respiratory dysfunctions. They usually have eight copies of the SMN2 gene [12].

For the diagnosis, genetic testing is required, with the detection of homozygous deletions of exon 7 of SMN1 in patients with the previously mentioned signs and symptoms being definitive. If genetic testing is available, electromyography and muscle biopsies are no longer mandatory [13]. The management of this disease must be multidisciplinary, especially in types 1–3, where respiratory failure may be a common complication. Additionally, a gastroenterologist and an orthopedic doctor must be part of the team [12].

The first oral drug approved for the treatment of types 1, 2, and 3 of this disease is risdiplam. It is an SMN2 pre-mRNA splicing modifier. That means that it modifies the function of the SMN2 gene, allowing it to function normally and generate full-length SMN protein. This leads to the amelioration of motor neuron functions [14].

It presents as a powder for oral solution, with each bottle containing 60 mg of risdiplam in 2 g of powder for oral solution. Each ml contains 0.75 mg of the active substance. It is administered based on body weight to patients 2 months of age or older, orally, once a day [15].

There were two pivotal clinical studies conducted to evaluate the efficacy and safety of this drug, FIREFISH and SUNFISH, that investigated the administration of this drug in patients with type 1, 2, or 3 [15]. The FIREFISH study is a two-part study in infants with type 1 SMA, with the first part analyzing the safety, pharmacokinetics, and pharmacodynamics of the drug and the second part focusing on the selection of the dose. 21 infants were enrolled in the first part of the study, with a median age of 6.7 months. After 12 months of treatment, 41% (7/17) of the patients were able to sit independently for a minimum of 5 s, and after a total of 24 months, another 3 children were able to sit, for a total of 10/17 patients who achieved this progress. A total of 90% (19/21) of the patients were alive and event-free after 12 months of treatment; 86% were able to feed orally; and after 33 months of treatment, 81% (17/21) were alive and event-free. There were three deaths during treatment and one 3.5 years after its discontinuation [16]. In the second part of the FIREFISH study, 41 patients were enrolled, with a median age of 5.3 months. After 24 months, 44% of patients were able to sit without support for 30 s; 80.5% were able to roll; 82.9% were event-free; and 92.7% were alive. A total of 85.4% had the ability to feed orally [15]. The SUNFISH study is also a 2-part study that investigates the efficacy, safety, pharmacodynamics, and pharmacokinetics of the drug in patients aged 2–25 years with SMA type 2 or 3. The following parameter was the change in the Motor Function Measure-32 score (MFM32), which assesses different motor functions; the higher the score, the better the motor abilities [15]. In the first part, 51 patients with SMA types 2 and 3 were included, with a significant improvement in the motor function measured by MFM32 after 1 year and a mean change from baseline of 2.7 points (95% CI: 1.5, 3.8), maintained after 2 years of treatment [15]. The second part, a randomized, double-blinded, placebo-controlled study, included 180 patients with a median age of 9 years, who received either risdiplam or a placebo. The mean change from baseline in the MFM32 total score after 12 months of treatment for those who were administered the active drug was 1.36 (95% CI: 0.61, 2.11), with a maintenance of the improvement after 24 months of treatment [15].

Another new orphan drug approved for the treatment of 5q SMA is nusinersen. It is the first approved drug to treat both adults and children with this condition. It is a synthetic anti-sense oligonucleotide that modifies the SMN2 gene in order to produce a full-length protein that works normally [13]. It is presented in vials of 5 mL, each containing 12 mg of the active substance, for intrathecal injection. There are 4 loading doses and a maintenance dose that must be administered once every 4 months [17]. Regarding its clinical efficacy and safety, there have been a few studies conducted in accordance with the age of onset. Study CS3B (ENDEAR), which is a phase 3, randomized, double-blind, sham-procedure controlled study, assessed the effect of the drug concerning motor function and survival. It analyzed 121 symptomatic infants, aged < 7 months. The results showed that 51% of the patients in the nusinersen group achieved a motor milestone, compared to the sham control group (0%) (*p* < 0.0001). A total of 25% of the nusinersen group required permanent ventilation compared to 32% in the sham-control group. Further, follow-up of these patients was made with the help of the study CS11 (SHINE), in which 89 patients treated with nusinersen were enrolled. Further improvement in motor milestones was noted, and 94% of the patients were alive at the time of this study. There is also an open-label Phase 2 study, CS3A, that showed that, from a total of 20 patients, 12 (60%) achieved a notable improvement in their motor abilities. At the end of the study, four patients needed permanent ventilation, and five died [17]. For the late-onset disease, there is the CS4 study (CHERISH), a phase 3, randomized, double-blind, sham-procedure controlled study. It gathered 126 patients, from whom 56.8% (95% CI:45.6, 68.1), *p* = 0.0006, compared to 26.3% (95% CI: 12.4,40.2), achieved an improvement in the Hammersmith Functional Motor Scale-Expanded scores after 15 months and 19.7% (95% CI: 10.9, 31.3), *p* = 0.0811, compared to 5.9% (95% CI: 0.7, 19.7), achieved new motor milestones at 15 months [17]. The improvement of motor function and survival rate is also supported by the CS2 and CS12 studies, 2 open-label studies that assessed patients over a period of 2 years and demonstrated the amelioration of the parameters. Additionally, progress has been noted in the six-minute walk test for the patients treated with nusinersen [17]. Those not eligible because of age or the number of SMN2 copies were enrolled in the CS7 (EMBRACE) study, which has two parts. All patients who received nusinersen were alive at the end of phase 1, except for one patient in the control group, and no patient required permanent ventilation. A total of 7 out of 9 patients (78%; 95% CI: 45, 94) were treated with nusinersen, and 0 out of 4 patients (0%; 95% CI: 0, 60) in the sham group met the criteria for a motor milestone response [17].

For adults, there are real-world clinical findings that support its beneficial effects. The study CS5 (NURTURE) observed the pre-symptomatic infants who were genetically diagnosed. 2 out of the 25 patients required respiratory intervention; all 25 patients achieved the milestone of being able to sit without support, and 22 (88%) were able to walk without assistance [17].

The third orphan drug approved for the treatment of this disease is onasemnogene abeparvovec. It is a recombinant adeno-associated viral vector that contains DNA capable of encoding SMN1 [13]. It presents as a 2 × 10^13^ vector genomes/mL solution for infusion, and it is administered as a single-dose infusion [18]. Its efficacy and safety were assessed in the AVXS-101-CL-303 (Study CL-303), a phase 3, open-label, single-arm study that analyzed 22 patients who received a single dose infusion of the drug. The results showed that 21 survived >10.5 months of age without the need for permanent ventilation, and 20 survived without other events to 18 months of age. There has been a 90.9% (95% CI: 79.7%, 100.0%) event-free survival at 14 months of age. 59.1% achieved the milestone of independent sitting for >30 s at 18 months of age [18]. These results are supported by the study AVXS-101-CL-101 (Study CL-101), which is a phase 1 study for those with SMA type 1. The study gathered 12 patients who received a single intravenous infusion of the drug. All patients were event-free at 14 months of age at 24 months of follow-up, and 10 out of 12 were able to sit independently [18]. The AVXS-101-CL-302 (Study CL-302) is also a phase 3, open-label, single-arm, single-dose study that included 33 patients with type 1 and type 2 SMA and 2 copies of SMN2. One patient died, and from the rest of the 32, 14 (43.8%) achieved the milestone of independent sitting at >14 months of age [18]. For the pre-symptomatic infants, there is the AVXS-101-CL-304 (Study CL-304) study, a phase 3, open-label, single-arm, single-dose administration of a single dose of medicine to newborns who are asymptomatic. A total of 2 cohorts were described, 1 consisting of 14 patients (with 2 copies of SMN2) and the other of 15 patients (with 3 copies of SMN2). In the primary cohort, all patients survived event-free to >14 months of age and did not require permanent ventilation. A total of 11 patients achieved independent sitting before 279 days of age, and 9 patients (64.3%) were able to walk alone. In the second cohort, there was a 24-month event-free survival without permanent ventilation. Between 9.5 and 18.3 months of age, all patients were able to sit independently, and 14 (93.3%) were able to walk [18].

#### 3.1.3. Lambert–Eaton Myasthenic Syndrome (LEMS)

Lambert–Eaton myasthenic syndrome (LEMS) is a rare disease of the neuromuscular junction that affects approximately 0.1 in 10,000 people in the European Union [19]. It is frequently associated with small cell lung cancer (SCLC), being an important marker for the early detection of this disease. It may also present as a non-paraneoplastic but rather primary autoimmune disorder, and it shows a certain association with the HLA-B8-DR3 haplotype [20]. Its pathophysiology consists of reduced acetylcholine (ACh) release in the synaptic space, with a normal quantity of ACh vesicles, a normal ACh presynaptic concentration, and normal postsynaptic ACh receptors. [21]. The ACh release into the nerve terminal is moderated by the calcium ion influx via the voltage-gated calcium channel (VGCC). It was demonstrated that in LEMS, the number of these channels is reduced due to the presence of IgG antibodies directed against them, and more precisely, against the P/Q type VGCC [20]. The clinical features consist of the weakness of the muscles, more often proximal than distal, with symmetrical implications, progressive, and the affection of the oculobulbar region in the advanced forms. The patients may experience dull pain and a feeling of stiff muscles, and they may present with areflexia but no significant muscle atrophy. Another clinical feature is the return of the tendon reflexes and muscle strength after repeated muscle contractions; this phenomenon being called postexercise or post-activation facilitation. Approximately 70% of patients may also present involvement of the cranial nerves, with the most common signs being ptosis and diplopia. 80% to 96% of the patients experience autonomic dysfunctions, such as dry mouth, erectile dysfunction, constipation, or orthostatic dysfunction. In the later stages of the disease, patients may experience respiratory failure [20]. A detailed clinical examination may raise the suspicion of LEM, but for a certain diagnosis, the presence of antibodies against P/Q-type VGCC along with specific electrodiagnostic studies is necessary. Screening for malignancy is also mandatory [20].

In addition to the symptomatic therapy available, the only drug that was considered to be efficient is amifampridine. It blocks the potassium channel, thus leading to the prolongation of the presynaptic nerve terminal membrane depolarizations and therefore enhancing the entrance of calcium and the release of ACh [22]. It is presented as tablets, each containing 10 mg of the active drug. It is administered in divided doses, three to four times a day. It starts with 15 mg a day, increasing to 5 mg every 4 to 5 days, to a maximum of 60 mg per day and 20 mg per single dose [23]. There are more small, randomized studies that evaluate the efficacy and safety of this drug. All these trials showed that amifampridine improved muscle strength in these patients. Amifampridine phosphate is effective and safe in the phase 3 clinical trial in LEMS. This phase 3, randomized, double-blind study included 38 adult patients randomly assigned to the active drug or placebo after a period of 7–91 days of treatment with amifampridine. The quantitative myasthenia gravis score was used to assess the results, and after 14 days of treatment, the patients from the amifampridine group showed better scores (mean difference 1.7 points, 95% CI: 0.0–3.4) and also a lower clinical progression of symptoms (6 versus 59 percent at day 8 and 12.5 versus 38 percent at 14 days) [24].

#### 3.1.4. Fabry Disease (α-Galactosidase A Deficiency)

Fabry disease is a disorder of lysosomal storage due to an error in the glycosphingolipid metabolic pathway. This leads to an excessive accumulation of globotriaosylceramide (GL-3), which is a glycosphingolipid metabolized in the lysosomes. Because of a deficiency of alpha-galactosidase A (alpha-Gal A), it accumulates in various cells and tissues. It appears to be an X-linked disease that may be determined by a pathological variant of the alpha-Gal A gene that is mapped on the long arm of the X chromosome. The most common implications are neurological, dermatological, renal, and cardiac [25]. Neurological afflictions include both the peripheral and the central nervous systems (CNS). The main cause of the implications of the CNS is the vasculopathy of the cerebral vessels, which leads to a high incidence of stroke. Some mild cognitive abnormalities were also described in relation to Gb3 accumulation. The affection of the peripheral nervous system is due to the damage preferentially of the small Aδ fibers, causing rare autonomic system failure and neuropathic pain [26]. The typical neuropathy described in this disease affects the small myelinated and unmyelinated neurons, being a symmetrical, length-dependent sensory polyneuropathy [27]. The diagnosis consists of genetic testing, with notable differences between men and women, and for the neurological affliction, nerve conduction studies, a quantitative sensory testing, quantitative sudomotor axon reflex test, and a skin biopsy are useful to determine the diagnosis [27].

Its treatment includes the orphan drug migalastat. It is a pharmacological chaperone that binds to certain forms of alpha-Gal A and stabilizes the mutant forms in the endoplasmic reticulum. This way, it enables suitable traffic to lysosomes. Once stabilized, the enzyme can perform efficient catabolism of GL-3. It presents as hard capsules that contain 123 mg of migalastat, and it is administered once every other day, at the same time of the day [28]. At the moment, there are 22 available European clinical trials concerning the efficacy and safety of this product [29]. Two phase three pivotal clinical studies and two open-label extension (OLE) clinical studies give pivotal data regarding the efficacy and safety of this product. The ATTRACT study compared the use of migalastat to enzyme replacement therapy (ERT) in 52 patients who were already on ERT. The effects on renal function were similar; the migalastat significantly decreased the left ventricular mass index compared to the ERT, and renal, cardiac, or cerebrovascular events appeared in 29% of patients treated with migalastat compared to 44% of the ERT group [30].

The FACETS study, also a phase 3 clinical study, evaluated this drug’s efficiency and safety in 50 patients who were either naïve to ERT or had stopped the ERT for at least 6 months. The results showed clinical benefit regardless of disease activity [31].

#### 3.1.5. Acute Hepatic Porphyria (AHP)

Porphyrias represent a group of diseases, being generated by defects in the enzymes mediating the pathway for the synthesis of heme, with an overproduction of it. Heme is an essential element in the function of hemoglobin and hepatic cytochrome P450. The neurological implications of porphyria are due to the excessive production of neurotoxic precursors [32]. The enzyme implicated in its pathology is porphobilinogen deaminase (PBGD), also known as hydroxymethylbilane synthase (HMBS). The appearance of symptoms also requires the induction of the delta-aminolevulinic acid (ALA) synthase (ALAS1), which is a rate-limiting enzyme for the synthesis of heme in the liver. Its activation leads to increased production and accumulation of ALA and PBG, which are considered to be toxic intermediates [33].

The neurological manifestations that can appear in AHP are due to a combination of central, peripheral, sensory, motor, and autonomic nervous system abnormalities; the exact mechanism for their appearance is not fully understood. One theory is the neurotoxicity of certain intermediates overproduced in the liver, and another is the limitation of the production of cellular hemoproteins due to heme deficiency, which affects neuronal and vascular functions [33]. The clinical manifestations vary, and some of the patients never develop syndromes. Usually, gastrointestinal or neurological symptoms such as abdominal pain, chest pain, or pain in the extremities are common due to abnormalities in the peripheral, central, or autonomic nervous systems. The duration of these attacks ranges from days to weeks, but this depends on the precipitating factors and treatment [32]. Peripheral neuropathy is usually both sensorial and motor, and the patients complain of pain in the extremities, along with numbness, paresthesia, or dysesthesias. The motor symptoms usually tend to appear later, after a prolonged attack. The motor weakness affects first the upper extremities and it can progress distally. Additionally, a prolonged attack may lead to the involvement of the cranial nerves, with bulbar paralysis and respiratory complaints [33].

Abdominal pain and other gastrointestinal symptoms may be due to the damage of the autonomic nervous system, the most common being tachycardia, but also cause hypertension, sweating, and tremors. This may be explained by the rise in the level of catecholamines during the attacks. Neuropsychiatric manifestations may also be reported, such as anxiety, restlessness, agitation, hysteria, hallucinations, or depression [33]. The involvement of the central nervous system may lead to seizures, and the diagnosis is very important because therapies such as phenytoin or valproic acid may aggravate the attack. Short-acting benzodiazepines, gabapentin, or magnesium can be used for the treatment of these seizures [32]. Cases of posterior reversible encephalopathy syndrome (PRES) and inappropriate antidiuretic hormone secretion (SIADH) have been noted in the literature. There may also be neuropathic bladder dysfunction that can cause dysuria, hesitancy, urinary retention, or incontinence [33]. This diagnosis, although it concerns a rare disease, must be kept in mind because early treatment can improve long-term outcomes and avoid future complications. It can be taken into consideration in patients who present with an acute episode of abdominal pain and/or neuropsychiatric symptoms. Of course, a full workup in this direction should be made after the usual investigations have excluded the most common causes. The screening of AHP PBG and total porphyrins should be measured using spot/random urine, and the results should be adjusted to creatinine levels [33].

In addition to the symptomatic treatment, a new therapy accepted by the FDA in 2019 and by the EMA in 2020, givosiran, is a compound that interferes with the production of the neurotoxic ALA and PBGD. It is a small interfering RNA that disrupts the normal functioning of the hepatic ALAS1 gene. It is administered as a subcutaneous injection once per month [33].

The drug’s efficacy and safety were evaluated in the ENVISION trial, which is a randomized, double-blind, placebo-controlled, multinational study. It summed up 94 patients with AHP, and they were randomized to either receive givosiran or a placebo for a period of 6 months. The efficacy was measured based on the rate of attacks that required hospitalization, urgent health care visits, or the necessity to administer hemin at home. Givosiran-treated patients had 70% (95% CI: 60%, 80%) fewer attacks than those on placebo [34].

#### 3.1.6. Non-Dystrophic Myotonias

The non-dystrophic myotonias are a group of neuromuscular diseases characterized by altered excitability of the cellular membrane. The difference with myotonic dystrophy is the absence of progressive muscular weakness and systemic characteristics. To this date, there are many different phenotypes based on the clinical appearance and the severity of the symptoms. The most common incriminated mutations in the pathogenesis of these diseases are in the genes of the major skeletal muscle voltage-gated chloride channel (CLCN1) and the sodium channel (SCN4A), with correlations between the genotype and the electrophysiological and clinical genotype [35]. These channelopathies lead to the disruption of the resting membrane potential and the normal processes of repolarization and depolarization, with subsequent hypo- or hyperexcitability or combined features of the two [36]. This group of disorders includes the classic myotonia congenita, paramyotonia congenita, hyperkalemic periodic paralysis with myotonia, and a group of sodium channel myotonias. Those with myotonia congenita, hence alterations in the CLCN1 gene, have classical symptoms of muscular involvement, with a classical aspect on examination (a delayed relaxation after the contraction) and improvement of myotonia with activity. The SCN4A group presents milder alteration of the muscles that permits the usual daily activities, but with the possibility of evolution to severe stiffness of the muscles or paralysis. Paramyotonia congenita presents with cold sensitivity, a “paradoxical” alteration of myotonia due to repetitive physical activity, and periodic weakness. Patients with hyperkalemic periodic paralysis may present with myotonia, but the main clinical feature is progressive weakness. Sodium channel myotonia patients do not manifest episodic weakness but may experience variable cold sensitivity [37]. The clinical examination shows myotonia as the main sign, and it can be elicited with muscle contraction or the percussion of the muscle. This manifestation has electrophysiological correlations. Patients may also present with periodic paralysis, manifested as flaccid weakness. For the diagnosis, the muscle biopsy is not necessary; the genetic testing shows either mutations in the CLCN1 or in the SCN4A channel gene. Depending on the form, the frequency of these mutations may vary. Electrophysiology testing may also bring valuable information regarding the form of the disease, based on the compound motor action potential through which the fiber excitability is being tested. The detection of the phenotype and genotype is very important to establish the most suitable treatment and to provide further genetic counseling [36].

Mexiletine, first used in the treatment of cardiac arrhythmias, was designated as an orphan drug once its efficiency in the treatment of non-dystrophic diseases was proven. It is a class I B antiarrhythmic treatment that has the potential to favor the inactivation of the sodium channels and, by doing so, stop the sodium ions from passing in and out of these channels. These movements dictate the relaxation and contraction of the muscles and present excessive activation in patients with myotonic disorders [38].

It presents as hard capsules, and it is important to mention that it has a multitude of adverse reactions that limit its use, including side effects involving the cardiovascular and central nervous systems, but also the gastrointestinal, musculoskeletal, and dermatologic systems, further detailed in Table 1. It has a black box warning according to the findings of the Cardiac Arrhythmia Suppression Trial that was published in 1989. It was concluded that the patients who were treated with antiarrhythmics of class I for asymptomatic, non-life-threatening ventricular arrhythmias post-myocardial infarction had greater mortality than those in the placebo group [39]. This drug’s efficacy and safety were evaluated in the MYOMEX study, a multi-center, double-blind, placebo-controlled, cross-over study that involved 25 patients, 13 with myotonia congenita and 12 with paramyotonia congenita. The symptoms had to affect at least 2 segments and have an impact on more than 3 daily activities. This study evaluated the score of stiffness severity, measured on the Visual Analog Scale (VAS), the changes in the quality of life measured by the individualized neuromuscular quality of life scale (INQoL), and the chair test (the time needed to stand up from a chair, walk around it, and then sit down again). The results showed that mexiletine decreased the stiffness score from 71.0 at baseline to 16.0 at the end of the treatment period versus the placebo group, in which the median VAS score did not modify. Moreover, the patients in the mexiletine group showed better results on the chair test compared to the placebo group. The INQqL scale was based on changes in the most common symptoms reported by the patients, namely weakness, muscular locking, pain, and fatigue. Mexiletine significantly improved these symptoms (the mean scores for locking decreased from 69.1 to 30.5 and for weakness from 63.4 to 30.5) compared to placebo (the mean scores remained the same). Additionally, the mean score for the overall quality of life was significantly improved with mexiletine [40].

#### 3.1.7. Hereditary Transthyretin-Mediated Amyloidosis (hATTR Amyloidosis)

Hereditary transthyretin amyloidosis (hATTR) is a rare, adult-onset, autosomal dominant, multisystem disease with heterogeneous manifestations. It affects the peripheral nervous system, leading to somatic and autonomic manifestations, but it may also involve the heart, kidney, and ocular vitreous. On average, the survival period is 10 years in untreated subjects. Based on the organ affected the most, there are two forms described, one being polyneuropathy (hATTR-PN) and the other being cardiomyopathy (hATTR-CM), although often the patients present with affection for both systems [41]. The disease is the result of several mutations in the gene that codes for transthyretin. This gene is located on chromosome 18, and the most common mutation incriminated in the pathology is a point mutation that leads to the replacement of valine by methionine at position 30 of the mature protein [41]. The main purpose of the TTR (previously known as pre-albumin) is the transport of thyroxine (T4) and retinol-binding protein. The main site of production is the liver. In this disease, the stability of TTR metamers is affected, leading to the misfolding of the proteins. Therefore, the damaged, dysfunctional TTR amyloid fibrils will be agreed upon and further deposited extracellularly. This condition should be differentiated from the age-related condition of senile amyloidosis [41]. Based on the type of mutation, the clinical phenotype can be dominated by cardiomyopathy, while others will have pure neurological manifestations. The initial manifestations can be fatigue or unintentional weight loss. The clinical picture depends also on the age of the debut. Patients may also present autonomic dysfunctions such as gastrointestinal symptoms (constipation, diarrhea, or their alternative appearances, orthostatic hypotension, bladder dysfunction, and erectile dysfunction). The cardiac implications may be diastolic dysfunction, conduction disorders, and congestive heart failure with preserved ejection fraction. Patients may also experience kidney and ocular involvement. The central nervous system can be involved, and cases of stroke, subarachnoid hemorrhage, hydrocephalus, cerebellar ataxia, spastic paresis, seizures, and dementia may appear [41]. The diagnosis is based primarily on the family history; a positive history is highly suggestive of the diagnosis and is usually established within a year from onset once the specific mutations are discovered. There are some problems establishing this diagnosis in the non-endemic regions, where the conclusions may take up to 5 years. If suspected, the next step is a tissue biopsy. With immunohistochemistry or mass spectroscopy, the confirmation of TTR as the precursor protein can be stated. The nerve biopsy has a sensitivity of 80%, but the periumbilical fat biopsy is also a method widely spread at present. Traditional nerve conduction studies are performed alongside tests to evaluate autonomic dysfunctions. Investigations for cardiac assessment are also necessary to stratify cardiomyopathy. The use of scintigraphy for the detection of the cardiac uptake of bone tracers may be an alternative to cardiac biopsies, and cardiac serum markers give important clues regarding the diagnosis. NT-proBNP and cardiac troponins (Tor I) have an important prognostic value. If the genetic diagnosis is confirmed, one must also undergo an ophthalmological evaluation. The evaluation of the kidney is also of great importance, with the monitoring of serum creatinine, proteinuria, or microalbuminuria, and the estimated glomerular filtration rate. The patients may have an altered body mass index because of gastrointestinal symptoms [41]. Until recently, the treatment of hATTR was made with tafamidis, a TTR stabilizer, and diflunisal, a nonsteroidal anti-inflammatory drug (NSAID), that also had the purpose of TTR stabilization.

More recently, two new orphan drugs were approved for the treatment of this disease: patisiran and inotersen. Both are efficient in the treatment of stage 1 or 2 polyneuropathy in hATTR amyloidosis. They represent gene-slicing therapies and function by reducing the production of both wild-type and mutant TTR by interfering with their mRNA (small interfering RNA-siRNA) [41]. Patisiran is an injectable product in the form of lipid nanoparticles, and it delivers siRNA to hepatocytes, which are the main site to produce TTR protein. Therefore, the TTR mRNA undergoes the process of catalytic degradation, and the serum protein levels are reduced [42]. The clinical study that evaluated the efficacy and the safety profile of this drug is APOLLO-B, a randomized, double-blind, placebo-controlled study that gathered 225 patients with a TTR mutation and polyneuropathy. They received either a placebo or patisiran (1:2), and all had premedication with corticosteroids, paracetamol, and H1 and H2 blockers [43]. Of these patients, 46% had stage I disease, meaning that the ambulation was not affected, and the neuropathy was mild with sensory, motor, and autonomic complaints. A total of 53% had stage II disease, meaning that they required assistance for walking, and the neuropathy was moderate, affecting not only the lower limbs but also the upper limbs and trunk. Half of the patients also presented with cardiac involvement, defined as a baseline left ventricular wall thickness >13 mm and no personal history of arterial hypertension or valve disease. The first endpoint that measured the efficacy of patisiran was the change in score in the modified neuropathy impairment score +7 (mNIS+7). It assesses motor, sensory, and autonomic polyneuropathy through the evaluation of motor strength and reflexes, a quantitative sensory test, nerve conduction studies, and postural blood pressure. The higher the score, the worse the affection. The patients presented an improvement in this score at 9 and 18 months, compared to placebo-treated patients. A total of 56.1% of patisiran-treated patients, compared to 3.9% of the placebo group, presented a better mNIS+7 score (*p* < 0.001). Additionally, 51.4% of the patisiran group had a benefit regarding quality of life, quantified with the Norfolk Quality of Life-Diabetic Neuropathy total score (QoL-DN), compared to 10.4% of the placebo-treated patients, first noted at 9 months of treatment. For the cardiac-affected patients, the patisiran group experienced a decrease in the LV thickness (LS mean difference: −0.9 mm [95% CI: −1.7, −0.2]) and longitudinal strain (LS mean difference: −1.37% [95% CI: −2.48, −0.27]), compared to the placebo group. Moreover, it caused the lowering of the NT-proBNP parameter [43].

Inotersen, the second orphan drug approved for the treatment of this condition, is an antisense oligonucleotide (ASO) that has the role of inhibiting the formation of TTR. It also accelerates mRNA degradation and therefore lowers the circulant levels of TTR. It is administered subcutaneously [44]. Its efficacy was observed in the NEURO-TTR study, a multicenter, double-blind, placebo-controlled that gathered 172 patients. There were three stages of the disease: stage 1 (patients who did not require assistance to walk), stage 2 (patients who required assistance to walk), and stage 3 (patients who necessitated a wheelchair). The mNIS+7 and QoL-DN were assessed. They showed a significant improvement in the inotersen-treated patients at 66 weeks, compared to the placebo. The differences in the least-squares mean change from baseline to week 66 between the 2 groups were −19.7 points (95% confidence interval [CI], −26.4 to −13.0; *p* < 0.001) for the mNIS+7 and −11.7 points (95% CI: −18.3 to −5.1; *p* < 0.001) for the QOL-DN score, independent of age, disease stage, or mutation type. The most frequent adverse reactions were glomerulonephritis and thrombocytopenia [45].

In 2022, Vutrisiran, a transthyretin-directed small-interfering ribonucleic acid, was approved for the treatment of polyneuropathy induced by hATTR [46]. It targets the TTR mRNA, and its final purpose is its catalytic degradation in the liver and, therefore, the reduction of serum TTR protein levels [47]. It is administered subcutaneously once every three months. It is important to acknowledge the possibility of a decrease in vitamin A levels; therefore, supplementation with vitamin A is recommended [48]. The study that evaluated its efficacy and safety is the HELIOS-A study, a randomized, open-label clinical study. The patients were either receiving Vutisiran or Patisiran, with a ratio of 3:1. The treatment period was 18 months, the efficacy of Vutisiran was assessed compared to a placebo group (the placebo arm from the APOLLO study), and the TTR serum reduction was compared with the Patisiran arm. The efficacy was measured depending on the changes in the mNIS+7 score after 18 months of treatment. Additionally, the changes in the QoL-DN score were evaluated. The benefit relative to placebo in these scores was similar, but the NT-proBNP levels were lower in the Vutisiran-treated patients. Vutisiran also showed an improvement in gait speed, nutritional status, and gait speed vs. placebo treatment [49].

#### 3.1.8. Refractory Generalized Myasthenia Gravis

Myasthenia gravis is an autoimmune chronic disease of the neuromuscular junction. It is mediated by B cells and associated with acetylcholine receptor (AChR) or muscle-specific receptor tyrosine kinase (MuSK) antibodies, which is characterized by muscle weakness and fatigability. The “general” term refers to the involvement of the ocular muscles, with a variable combination of arms, legs, and respiratory muscles. The refractory form of this illness (gMG) is defined by the lack of response to conventional treatment [50]. The cardinal clinical characteristics are the fluctuating weakness and fatigue of the affected muscle groups. In the beginning, there are usually described periods of amelioration, that tend to disappear after a long evolution. Most of the patients present with primary ocular symptoms, such as ptosis or diplopia; approximately 15% present with bulbar symptoms such as dysarthria, dysphagia, and fatigable chewing; and a minority of patients present only with weakness of the limbs [51]. The diagnosis is made based on the clinical history and the typical signs of muscle weakness. Paraclinical investigations include serological tests for autoantibodies and electrophysiologic studies [51]. The therapies used for the treatment of this disease are the symptomatic medications with the role of raising the amount of acetylcholine available at the neuromuscular junction, the chronic immunotherapies that target the underlying immunological imbalance, the acute immunotherapies, such as plasma exchange and intravenous immune globulin, and lastly, surgical treatment with thymectomy [52].

Eculizumab is a humanized monoclonal antibody, also discussed in the section about NMOSD in this paper. Its capacity to inhibit the formation of the C5b-induced membrane attack complex is very useful considering the role of complement-mediated injury to the postsynaptic membrane in the pathogenesis of this disease. The first study that demonstrated the benefit of this drug in gMG was made in 2013, on a total of 14 patients with severe and refractory gMG, and showed a significant improvement in the Eculizumab-treated patients’ group compared to the placebo group [53]. The REGAIN study is another trial, a multicenter one, made on 125 patients with gMG that also favored Eculizumab over placebo, concerning the quality of life, the quantitative MG scale score, and the rate of exacerbations. In its extension made on 117 patients, treated with 1200 mg of Eculizumab every 2 weeks for 3 years, the functional improvement was maintained over time, and the exacerbations were 75% lower than those in the 1-year pre-study. It also lowered hospitalization rates and the need for rescue therapies [54]. It is of crucial importance to be aware of the possible increase in life-threatening Neisserial infections, and therefore, the patients should be immunized with meningococcal vaccines two weeks before the administration of the first dose. Additionally, oral antibiotic prophylaxis may be taken into consideration given the high risk of infection [55].

#### 3.1.9. Duchenne Muscular Dystrophy

Duchenne muscular dystrophy (DMD) is a dystrophinopathy caused by mutations in the DMD gene, a gene located on the X chromosome, which is responsible for coding the dystrophin. Dystrophin is a component of a large associated glycoprotein complex, and its role is to provide reinforcement to the sarcolemma and to stabilize the glycoprotein complex and, by this function, shield it from degradation. If dystrophin is absent, the complex is delocalized from the membrane, causing membrane fragility and increased permeability. This will in turn cause an increased calcium influx with the activation of the calcium-dependent proteases and the subsequent digestion of the cytoskeletal and myofibrillar proteins. Additionally, the production of neuronal nitric oxide synthase is affected. This is a protein responsible for producing nitric oxide and, therefore, regulating vasodilation and blood flow to the muscles. Most frequently, there are deletions of one or more exons, but gene duplications can also be seen [56].

The most important symptom is weakness, which usually occurs between two and three years of life. The first affected muscles are proximal, before the distal, and the lower extremities are involved first, before the upper extremities. The children can present pseudohypertrophy of the calf and occasionally of the quadriceps muscles, lumbar lordosis, hypotonia, and hypo- or areflexia. Because of the frequent fallings, these children usually have fractures. Among these patients, it is also described as a primary dilated cardiomyopathy with conduction abnormalities and a variety of arrhythmias. There may also be a mild degree of cognitive impairment or a delay in general development. The phenotype of the disease depends mostly on the quantity of residual dystrophin in the muscles, with higher levels being correlated with milder clinical pictures [56]. The diagnosis should be suspected when a young child with a positive maternal family history presents troubles in acquiring walking milestones. Additionally, if there is no evidence of family history, DMD can be suspected if the child is not walking by the age of 16 to 18 months or if the Gower’s sign, toe walking, and calf hypertrophy are present. Biologically, there is an unexplained increase in the transaminase values with normal liver function. In order to evaluate these patients, the creatine kinase level should be dosed at high levels, along with the clinical picture, which is highly suggestive. Molecular genetic testing can give a definite diagnosis; the muscle biopsy has been replaced in recent years by this more rapid and facile method. If there is no mutation that is causative for this disease, a muscle biopsy should be performed [56].

A new orphan drug, ataluren, was approved in Europe for the treatment of DMD. In 10–15% of the cases, a nonsense mutation in DNA causes a premature stop codon within an mRNA, resulting in the termination of the translation before the full-length protein is generated. The role of ataluren is to enable the ribosomal read-through of the premature stop codons, therefore restoring the full-length dystrophin protein. It is a drug suitable for patients aged 2 or older with DMD caused by nonsense mutations [57]. A certain clinical benefit has not yet been fully established. Clinical efficacy and safety were assessed in two clinical trials. The evaluated parameters were the change in 6-min walk distance (6MWD) at week 48, the time to a persistent 10% worsening in 6MWD, the change in time to run or walk 10 m at week 48, the time to climb 4 stairs at week 48, and the time to descend 4 stairs at week 48 [57]. The first study evaluated 174 patients, some receiving 40 mg/kg/day of ataluren, some 80 mg/kg/day of ataluren, and some receiving a placebo. At week 48, the patients receiving 40 mg/kg/day of ataluren had a 12.9 m decline in the 6MWD compared to the placebo group, which presented a 44.1 m decline. There was no difference noted between the 80 mg/kg/day of ataluren and the placebo. Additionally, fewer patients who received ataluren 40 mg/kg/day worsened in 6MWD over 48 weeks compared to placebo. Concerning the timed function tests, the ataluren patients showed smaller increases in the time it takes to walk or run 10 m, climb 4 stairs, and descend 4 steps, relative to placebo. Additionally, no significant difference was seen between the 80 mg/kg/day of ataluren and the placebo group [58]. The second study evaluated 230 male patients, who received ataluren 40 mg/kg/day or a placebo. The patients in the ataluren group presented clinical benefits versus placebo, demonstrated by lower changes in the 6MWD from baseline to week 48. Moreover, the ataluren patients presented a lower decline in muscle function, as seen in the time to run or walk 10 m, climb 4 steps, and descend 4 stairs [58].

The actual standard of care for the treatment of patients with DMD is corticotherapy, although this is still under debate due to its important side effects and long-term complications. However, there are many known benefits of this therapy, including the improvement of muscle strength, motor function, pulmonary function, the development of cardiac complications, and the overall effect of increasing survival. These effects are mainly due to their immunosuppressive and anti-inflammatory properties [59]. Deflazacort, a drug approved for the treatment of DMD in 2017 by the FDA and in 1993 by the EMA, is a synthetic glucocorticoid that has certain structural characteristics different from prednisone/prednisolone, thus helping with the minimalization of the classical corticoid’s adverse reactions. It does not have a sodium-retaining activity, nor does it lower the interference with carbohydrate metabolism [59]. It is a prodrug activated by esterases that binds to plasma proteins and blood cells, reaching a maximum plasmatic level in 1.5 to 2 h. The lower lipid solubility assures a lower passage through the blood-brain barrier, minimizing the adverse reactions to the behavior. It also has a lower impact on calcium metabolism, resulting in a lower risk of osteoporosis or osteopenia. The principal action is interfering with multiple gene expression pathways, with an elevation in the myogenesis-responsible genes and a reduction of the inflammatory ones [60]. The first study that assessed this drug dates from 1995 and was a phase III randomized, placebo-controlled, double-blinded trial that evaluated the strength in a sample of 196 boys. There were 4 groups: one treated with a dose of 0.9 mg/kg/d of deflazacort, one with 1.2 mg/kg/d of deflazacort, one with 0.75 mg/kg/d of prednisone, and one with a placebo. The first evaluation of strength after 12 weeks showed an improvement that was statistically significant for the first 3 groups compared to the placebo group. Additionally, there has been an improvement in the time from supine to standing and in the time to run 30 feet or climb 4 stairs compared to the placebo. After another 40 weeks, the group treated with deflazacort showed improved muscle strength compared to the prednisone group (LS mean 0.29, *p* = 0.044, 95% CI: 0.08–0.49). The group treated with prednisone experienced more adverse reactions compared to the group treated with deflazacort [60]. A newer study that gathered subjects from 2013 to 2016 and was also a double-blind, randomized trial that included boys aged from 4 to 7 years compared the efficacity and the adverse reactions from 3 different treatment regimens (corticosteroid alternatives). It showed that a daily administration of prednisone or deflazacort was more efficient than an intermittent administration, the last two having no great differences in the primary outcome. The results were mainly based on the differences in the velocity of rising from the floor. The forced vital capacity showed no great modifications. The results of this study support the daily use of a corticotherapy regimen versus an intermittent one [61].

Although most of the existing therapies aim at palliation, new medications targeting the use of nucleic acid-based drugs with exon skipping and the fix of the defective DMD gene tends to give a curative attitude. One of these drugs is eteplirsen. Its use is based on the observation that an in-frame mutation, with the production of a truncated version of dystrophin, gives a better clinical outcome compared to an existing out-of-frame mutation [62]. Eteplisren is a synthetic nucleic acid analog and, more precisely, a 30-nucleotide phosphorodiamidate morpholino oligomer. Synthetic nucleic acid analogs interfere with pre-mRNA splicing, leading to the exclusion of an exon from the final transcript. It causes exon 51 of DMD to be skipped during splicing, producing a shortened functional dystrophin protein. It is therefore beneficial in patients with deletions that end at exon 50 and start at exon 52 (14% of all DMD patients), being the single group of patients in which the skipping of exon 51 is applicable. Its efficacity was tested in four major clinical trials, the NCT02255552 being the one required by the FDA for final approval. It is a phase II open-label study, with an 80-patient cohort, treated with eteplisren or untreated. The efficacity in all four studies was measured based on the amounts of dystrophin produced and the effects on ambulation. The NCT02255552 study showed an improved dystrophin expression, independent of the dose given, with those treated with 30 mg/kg/week having an improvement of 22.9% in the dystrophin-positive fibers compared to the values before treatment (*p* ≤ 0.002), an effect not noted in the placebo-treated group. The NCT01396239/NCT01540409 studies assessed functional outcomes in terms of walking and respiratory function, showing a significant improvement in the 50 mg/kg/week treated patients (*p* ≤ 0.001). A persistent challenge concerning this medication is rapid clearance, with a low target tissue uptake [63].

Another synthetic antisense oligonucleotide approved for the treatment of DMD is golodirsen, whose target is exon 53, making it a suitable therapy for patients presenting with a mutation correctable by the skipping of this exon. It is administered intravenously once per week and has a low drug interaction potential. Its safety and efficacity were evaluated in the NCT02310906 clinical trial, a phase I/II, 2-part, multicenter trial. The first part, a randomized, double-blind, placebo-controlled phase, compared the patients receiving golodirsen 4–30 mg/kg/week to those receiving a placebo. Phase II evaluated the evolution over 168 weeks. The results showed a functional amelioration in golodirsen-treated patients, with a smaller deficit of walking after 3 years and a loss of ambulation in 9% versus 26% of the patients (*p* = 0.21) [64].

Another drug that targets exon 53 is viltolarsen. Its benefits are shown in a phase 2 clinical trial NCT02740972, which demonstrates a significant drug-induced dystrophin production and an important improvement in the functional tests, such as the time to stand from supine, run or walk 10 m, and the 6-min walk test [65].

Casimersen, a drug with a similar mechanism, targets patients with a mutation that can be addressed by skipping exon 45 of the pre-mRNA, with the before-mentioned result of a truncated but functional dystrophin protein [66]. It is administered once a week as an intravenous infusion. Its clinical safety and efficacy were evaluated in a clinical multicenter phase I/II study, with a first phase of 12 weeks and a second one of 132 weeks. The patients received escalating doses of casimersen and a placebo. It is also being investigated in the present with the ESSENCE trial, a phase 3, multicenter, placebo-controlled study, that will further analyze the efficacy and limitations of this drug [67].

#### 3.1.10. Narcolepsy with or without Cataplexy

Narcolepsy is a disease characterized clinically by daytime sleepiness, with or without cataplexy, hallucinations, and sleep paralysis. There are two types of narcolepsies: type I, which is the type with cataplexy, and type II, which is the type without cataplexy [68]. Cataplexy refers to muscle weakness that is triggered by emotions and is transitory. It usually affects the muscles of the face, neck, and knees. The first manifestations can be in the facial muscles, with ptosis and hypotonia of the mouth, with the interruption of smiling or other facial expressions. Bilateral paralysis may also occur, and the consciousness usually remains intact [69]. Its etiology is linked to the loss of orexin-A and orexin-B, neurotransmitters produced in the lateral hypothalamus, with an excitatory effect on the OX1 and OX2 receptors from the postsynaptic neurons. They increase the activity of the locus coeruleus, raphe nuclei, and tuberomammillary nucleus, inhibiting rapid eye movement (REM) sleep. It is because of this dysfunction that patients may present with cataplexy or hypnagogic hallucinations in a wakeful state. Most commonly, the narcolepsy type I people are the ones with an orexin deficit. In their case, an increased number of histamine-producing neurons and a reduction in the corticotropin-releasing hormone-producing neurons have been observed [70]. The cause of narcolepsy type II is still under debate, with these patients presenting normal orexin-A CSF levels. Approximately one-quarter of them have low orexin-A levels, but the causes may vary [71]. Another factor implicated in the appearance of this disease is environmental pollution, with studies showing an existing toxin that targets and destroys the hypocretin-producing cells in humans with a certain HLA profile (DQB1*0602) [72]. This observation is consistent with the new focused approach regarding the linkage between neurological diseases (vascular but also autoimmune) and air pollution [73].

Genetic factors may also play an important role, with several haplotypes being described in connection to this disease, but also environmental factors are incriminated in the development of it [74]. Being associated with the HLA DQB1*0602 haplotype, it is stated that the orexin neurons may also be killed in a selective way by an autoimmune process. One of the possible triggers is streptococcal pharyngitis [75]. Secondary cases have also been described, with the cause being several lesions in the posterior hypothalamus and midbrain. Additionally, the orexin neurons may be affected by tumors, vascular malformations, strokes, and inflammatory processes [76,77]. Usually, patients with secondary narcolepsy also have other neurological symptoms, such as neurological deficits and cognitive, endocrine, or motor abnormalities. The clinical picture is dominated by chronic daytime sleepiness, with or without cataplexy, hypnagogic hallucinations, and sleep paralysis. The main complaint is excessive sleepiness [78]. The diagnosis is based on the clinical features all the patients complain of daytime excessive sleepiness. When narcolepsy is suspected, a polysomnogram and a multiple sleep latency test are useful [79]. If the tests are inconclusive, the levels of orexin-A in CSF may be measured [80]. HLA testing is not routinely performed [81].

In addition to the changes in lifestyle and the existing pharmacotherapy, a new drug, pitolisant, is available for the treatment of this condition. It is an oral histamine H3 receptor inverse agonist that has the function of improving daytime sleepiness and reducing cataplexy. It is usually used for people who do not respond to or do not tolerate other drugs. Other medications available are modafinil, which is also the first line of therapy, armodafinil, methylphenidate, amphetamines, solriamfetol, oxybates, and antidepressants. Pitolisant is an oral drug that increases the levels of acetylcholine, noradrenaline, and dopamine in the brain. There are more studies concerning its safety and efficacy, the most important being Harmony I and Harmony CTP. Additionally, long-term safety was evaluated in the Harmony III study. The Harmony I study compared pitolisant with placebo and modafinil and included 94 patients. 60% of the patients reached 36 mg of pitolisant daily. The scales for the evaluation of its efficacy were the Excessive Daytime Sleepiness (EDS) and Epworth Sleepiness Scale (ESS). The pitolisant group had much better results than the placebo group (mean difference: −3.33; 95% CI [−5.83 to −0.83]; *p* < 0.05), but it did not have significantly different results compared to the modafinil group (mean difference: 0.12; 95% CI [−2.5 to 2.7]). It also showed a benefit for cataplexy attacks, with the frequency being notably smaller in the pitolisant group (−65% vs. −10%) [82]. Harmony CTP is a double-blind, randomized, parallel-group study that also compared pitolisan with a placebo and targeted its action in cases with a high frequency of cataplexy attacks. It measured the change in their number between week 2 and week 4 of stable treatment, assessing a total of 105 patients. The Weekly Rate of Cataplexy episodes had a 64% decrease in the pitolisant group, with a half reduction in the placebo group. The effect size of the drug compared to the placebo was summarized with the ratio rate rR(Pt/Pb), rR = 0.512; 95% CI [0.435 to 0.603]; *p* < 0.0001). Additionally, ESS decreased more in the pitolisant group compared to the placebo group [83]. Harmony III assessed its efficacy over a period of 12 months, with an extension to 5 years, including 102 patients. After one year, the ESS score decreased to 3.62. The study showed a significant improvement in symptoms such as sleep attacks, sleep paralysis, cataplexy, and hallucinations [84].

#### 3.1.11. Pompe Disease

Pompe disease, or acid alpha-glucosidase (GAA) deficit, is an autosomal recessive lysosomal storage disease, or glycogen storage disease type II (GSD II), that affects 1 in 40,000 newborns. It is characterized by an excessive accumulation of glycogen in all body tissues, with the main cause being the alteration in the degradation of glycogen [85].

The genetics of this disease is explained by the pathologic variant of the gene responsible for the lysosomal acid alpha-1,4-glucosidase (GAA). The accumulation of glycogen in the lysosomes and cytoplasm, apart from the destruction of the tissues, may also cause damage to the vesicle systems that are linked to lysosomes (glucose transporter 4) [86].

The clinical picture is characterized by a classical infantile onset with hypertrophic cardiomyopathy and muscle hypotonia, but it can also present as a late-onset disease without cardiac complications. The late onset is characterized by a skeletal myopathy with a possible course toward respiratory failure and death in the second or third decade of life. The muscular affection presents as a progressive weakness, predominantly proximally, particularly in the hip flexors. They can also be associated with arteriopathy dilatative, dissection of the carotid arteries, or basilar artery dolichoectasia. Upper and lower gastrointestinal symptoms have also been noticed [87].

The diagnosis is usually clinically suspected in a child presenting with important hypotonia and cardiac insufficiency. Laboratory modifications, such as elevated creatine kinase, lactate dehydrogenase, and aspartate aminotransferase levels, may be observed. In patients with weakness in a limb-girdle distribution, the late-onset form must be suspected. The changes observed on the electromyogram are characteristic. For the diagnostic, the measurement of the GAA enzyme in white blood cells or dried blood spots may be useful. The confirmation of the diagnosis is made with the help of gene sequencing, with two pathogenic variants in trans in the GAA gene being the diagnostic proof. The most common mutation is the C.-13–13T.G splice mutation. A muscle biopsy with the subsequent measurement of the GAA activity can be made, but this is a less used method due to its invasiveness. The GAA deficiency can also be tested in newborn screening panels [88].

Without treatment, survival is very limited, making early diagnosis and medication very important.

The first choice of treatment for GAA deficiency is alglucosidase alfa, an enzyme-replacing therapy. It is a recombinant form of the GAA that has a role in glycogen cleavage by binding to the mannose-6-phosphate receptors on the cell surface, being internalized, and being transported to lysosomes, where it enhances glycogen degradation. It is administered intravenously every two weeks. It is a drug recently approved by the FDA for the treatment of this disease in patients older than 1 year of age with late-onset deficiency. For this form of the disease, there is a multicenter trial that shows improved motor (measured by the 6-m walking test—6 MWT) and pulmonary function with the administration of this treatment compared to a placebo. The results show a differential treatment effect of 28.1 m in the 6 MWT and 3.4% in vital capacity [88]. The alglucosidase alfa can also be administered in infantile-onset disease, the studies showing a reduction in the risk of death, invasive ventilation, and cardiac and motor function. The pivotal, randomized, open-label, historical, controlled clinical trial that assessed its efficacity and safety for 18 infantile-onset patients showed that after 52 weeks of treatment, 15 of the 18 patients were alive and free of invasive ventilatory support, compared to 1 of 42 patients from an untreated historical cohort. At the end of the study (119 weeks), 14 of 16 patients were alive, and 9 of 16 were alive and free of ventilatory support. Therefore, a prolonged survival compared to the historical cohort was observed. Additionally, 7 of the 18 patients have made motor development gains, walking independently at the last evaluation. The use of this medication for late-onset disease is also supported by 4 important clinical cases, that show a conclusion an improvement or a stabilization of the motor function and of the pulmonary function, up to 5 years in the study [89].

## 4. Practical Steps

Analyzing the studies available and the new treatment options discovered in the last few years, it is safe to affirm that modern medicine has many more resources now to treat the diseases that once were thought to be incurable and only treated those that were symptomatic or palliative. In view of the international context of Europe, however, the accessibility is still not very facile for the patients, and remains very heterogeneous, depending on the healthcare organizations and drug reimbursement [90]. The first country to establish firm regulations for the development of orphan drugs was the United States, in 1983, with the introduction of the US Orphan Drug Act. In the European Union, the official status of the orphan drug did not exist until 2000. The status of these drugs and diseases is based on their cost of production and contribution to innovation, and in Europe it is based on equality, meaning that the patients suffering from diseases for which no treatment has been studied enough should still benefit from the same quality of therapeutic measures [91]. One problem that is brought to attention is the cost-effectiveness of the development of these drugs. Based on the severity of the disease, the evidence of life-quality improvement, and the vital risk of the disease, the national healthcare system should decide whether it should pay the prices for these drugs or not. The randomized control trials and the observational studies, some of which are quoted here, are meant to bring evidence-based arguments in favor of establishing these new therapeutic measures [92]. As we have evidenced in this paper, each orphan drug has at least one clinical randomized trial that has studied the impact on different parameters (quality of life, rate of exacerbations, clinical outcome) and supports its benefit, bringing arguments in favor of its development and its further foundation. Some consider that the analysis based only on the improvement of the quality-adjusted life years is not sufficiently sensible, and the treatment should be considered, especially if it may arrest a disease process or preserve life for patients with a very poor prognosis [93]. However, as already evidenced in this paper, drugs with common indications can also receive the status of an orphan drug depending on the disease they target, allowing pharmaceutical companies to use the maximum amount of public funding while minimizing their financial risk.

## 5. Conclusions

After analyzing a few of the existing orphan drugs that were relatively recently introduced in medical practice and the clinical studies that support their use, it is safe to assume that future development and an even more intense founding can bring many changes, not only in patients’ lives but also in the medical scene. So far, the humanity and professionalism of the pharmaceutical companies, along with the constant support of the patient’s associations for rare diseases, have led to the discovery of new treatments and future useful findings, giving a valuable direction for diseases that until a few years ago were thought to be incurable.

## Figures and Tables

**Table 1 jpm-13-00420-t001:** Principal orphan diseases and their treatment.

Disease	Drug	FDA Approval	EMA Approval	Most Common Adverse Reactions
NMOSD	Satralizumab	14 August 2020	24 June 2021	Common cold, headache, upper respiratory tract infections, gastritis, rash, joint pain, extremity pain, tiredness, and nausea.
aHUS; gMG; NMOSD	Eculizumab	27 July 2019	20 June 2007 26 August 2019	Upper respiratory infection, common cold (nasopharyngitis), diarrhea, back pain, dizziness, influenza, joint pain (arthralgia), and contusion.
SMA	Risdiplam	7 August 2020	26 March 2021	Fever, diarrhea, rash, commonly seen in infants, respiratory tract infection, pneumonia, constipation, and vomiting.
5q SMA	Nusinersen sodium	23 December 2016	30 May 2017	Low blood platelet count and kidney toxicity.
5q SMA	Onasemnogene abeparvovec	24 May 2019	18 May 2020	Hives, difficulty breathing, swelling of the face, lips, tongue, or throat; jaundice; easy bruising; unusual bleeding; purple or red spots under the skin.
LEMS	Amifampridine	28 November 2018	23 December 2009	Tingling around the mouth, tongue, face, fingers, toes, and other body parts, upper respiratory infection, stomach pain, nausea, diarrhea, headache, increased liver enzymes, back pain, high blood pressure, and muscle spasms.
Fabry disease	Migalastat	10 August 2018	26 May 2016	Headache, nasal and throat irritation (nasopharyngitis), urinary tract infection, nausea, and fever.
AHP	Givosiran	20 November 2019	2 March 2020	Life-threatening allergic reactions, serious liver and kidney injury, and injection site reactions.
Non-dystrophic myotonic disorders	Mexiletine hcl	1 October 2019	18 December 2018	Lightheadedness, tremors, coordination difficulties, dizziness, chest pain, palpitations, and dyspnea.
hATTR amyloidosis	Patisiran sodium	10 August 2019	26 August 2018	Infusion-related reactions and decreased vitamin A levels.
hATTR amyloidosis	Vutrisiran	June 2022	July 2022	Dyspneea, arthralgia, pain in the extremities, reaction at the injection site, and an increase in the levels of phosphatase alkaline.
hATTR	Inotersen	5 October 2018	10 July 2018	Nausea, headache, pyrexia, peripheral edema, chills, vomiting, anemia, and thrombocytopenia.
Duchenne muscular dystrophy	Ataluren	Not yet approved	31 July 2014	Vomiting, diarrhea, nausea, headache, upper abdominal pain, flatulence, hypertriglyceridemia, epistaxis, limb pain, hematuria, and enuresis.
Duchenne muscular dystrophy	Deflazacort	February 2017	October 1993	Cushingoid appearance, increase in weight and appetite, superior respiratory tract infections, cough, pollakiuria, hirsutism, obesity, and nasopharyngitis.
Duchenne muscular dystrophy	Eteplirsen	September 2016	September 2018	Headache, fever, falls, abdominal pain, cough, and nausea.
Duchenne muscular dystrophy	Golodirsen	30 August 2019	12 December 2019	Headache, fever, falls, abdominal pain, nausea, cough, and nausea.
Duchenne muscular dystrophy	Casimersen	25 February 2021	22 February 2021	Upper respiratory tract infection, cough, fever, headache, pain in the joints, mouth, throat, and ear, nausea, and dizziness.
Duchenne muscular dystrophy	Viltolarsen	Not yet approved	August 2020	Upper respiratory tract infection, injection site reaction, cough, and pyrexia.
Narcolepsy/ cataplexy.	Pitolisant	14 August 2019	31 March 2016	Heart rhythm problems, difficulty sleeping, nausea, and feelings of worry.
Pompe disease	Alglucosidase alfa	30 August 2006	28 April 2006	Skin rash, diarrhea, constipation, vomiting, pain in the arms or legs, and infusion site reactions.

Caption: NMOSD—Neuromyelitis optica spectrum disorders; SMA—Spinal muscular atrophy; LEMS—Lambert–Eaton myasthenic syndrome; AHP—Acute hepatic porphyria; hATTR amyloidosis—Hereditary transthyretin-mediated amyloidosis; aHUS—Atypical hemolytic uraemic syndrome; and gMG—Refractory generalized myasthenia gravis.

## Data Availability

Not applicable.

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
