# Peer review of "Orphan Drugs in Neurology—A Narrative Review"

_jpm, 2023, doi:10.3390/jpm13030420_

Round 1

Reviewer 1 Report

This is a fairly traditional review of orphan diseases and approaches to their treatment. My main criticism is related to the approach taken by the authors. I don't quite understand why they limited themselves to searching for rather narrow keywords. This affects the completeness of the presented material. For example, I would like to focus on Duchenne muscular dystrophy. It is a highly complex pathology affecting many systems, including cellular ion homeostasis, inflammation, fibrosis, metabolic dysfunction, and oxidative stress. For each such violation, specific pharmacological preparations (including those based on adenovirus particles) were selected. But the authors described only ataluren, which manifests itself rather contradictory, although, undoubtedly, the authors were right in citing it as an example. However, numerous other approaches should also be mentioned. This also applies to other considered pathologies. It should be noted that authors often use an encyclopedic approach, which is unacceptable. In addition, the review should be interesting for a potential reader; for this, the authors should supplement it with high-quality informative illustrations.

Reviewer 2 Report

This is an extensive review of orphan drugs utilized in neurological diseases. 

Major points

-       A new drug has been recently approved by FDA and EMA for hATTR amyloidosis, vutrisiran. Cite it together with the results of the clinical trial. 

-       Atypical haemolytic uremic syndrome is not a neurological disease. It has to be omitted.

-       Duchenne muscular dystrophy:

“the complex is digested by the proteases”. This is wrong. When dystrophin is absent, glycoprotein complex is delocalized from the membrane leading to membrane fragility and increased membrane permeability. Structural weakness and tears in the sarcolemma and consequent fail in membrane repair cause influx of calcium-rich extracellular fluid. When the calcium-buffering capacities of the sarcoplasmic reticulum, the mitochondria and intracellular calcium-binding proteins are reached, calcium-dependent proteases are activated leading to digestion of cytoskeletal and myofibrillar proteins.

Last sentence of the first paragraph (“it is an X-linked inherited disease ….) is a repetition of that said before. Omit it.

Indicate the percentage of DMD patients with premature stop codon, who can be treated with ataluren.

Deflazacort has been approved for DMD. Cite it.

Exon-skipping ASOs have been approved by FDA for DMD: eteplirsen, golodirsen, casimersen, viltolarsen.

-        Pompe disease has to be included in the review.

Minor points

-       Introduction: It is not unusual for a family doctor (or GP) to encounter less than one such case per year.

-       Spinal muscular atrophy: Start a new paragraph with the sentence “For the diagnosis genetic testing is required, the detection ….”

-       Acute hepatic porphyria, last sentence: “Givosiran treated patients ….”

-       Non-dystrophic myotonias, last paragraph, second line: “… reactions, that limit its use, … “

-       hATTR amyloidosis:

line 4: “On average, the survival period is 10 years in untreated subjects

“gastrointestinal symptoms (constipation, diarrhea, or their alternative appearance, orthostatic hypotension, bladder, and erectile dysfunction).

Cite periumbilical fat biopsy which is much more used than nerve biopsy now.

“They represent gene-splicing therapies, and function by reducing the production of both wild-type and mutant TTR, by interfering ….”.

“Therefore, the TTR mRNA undergoes the process of catalytic degradation, and the serum protein levels are reduced”.

Efficacy instead of efficacity.

“Inotersen, the second orphan drug approved for the treatment of this condition, is an antisense oligonucleotide (ASO), that has the role of inhibiting the formation of TTR. It also accelerates mRNA degradation, and therefore lowers the circulant levels of TTR".

-       Myasthenia gravis: cite anti-MUSK antibodies.

Duchenne muscular dystrophy:

Modify as following: “The diagnosis should be suspected when a young child, with a positive maternal family history …”

Modify: “Biologically, there is an unexplained increase in the transaminase values, with normal liver function”.

Round 2

Reviewer 1 Report

The authors have improved the presentation of the manuscript. The work may be accepted. 

Author Response

Thank you for your time and your valuable comment!

Reviewer 2 Report

Still some changes are needed.

- Atypical haemolytic uremic syndrome is still mentioned in the key-words

- Pompe disease is not mentioned in the key-words

- Duchenne M.D.: The sentence "The calcium regulation may also be affected, with a pathological entrance of calcium into the muscle fibers. Excess can activate calpains and they promote muscle proteolysis." has to be omitted. The matter is already said in the sentence above “This will in turn cause an increased calcium influx, with the activation of the calcium-dependent proteases and the subsequent digestion of the cytoskeletal and myofibrillar proteins.”

-       DMD: Cite in the paragraph about deflazacort recent paper by Guglieri et al. Effect of Different Corticosteroid Dosing Regimens on Clinical Outcomes in Boys With Duchenne Muscular Dystrophy: A Randomized Clinical Trial. JAMA. 2022 Apr 19;327(15):1456-1468.

-       hATTR: Therefore, the TTR mRNA undergoes the process of catalytic degradation, and the serum protein levels are reduced [42].

-       Myasthenia gravis: The first sentence “Myasthenia gravis is an autoimmune chronic disease of the neuromuscular junction, mediated by anti-MUSK antibodies targeting the acetylcholine receptor (CchR), characterized by muscle weakness and fatigability.” Is wrong.  

Change in: “Myasthenia gravis is an autoimmune chronic disease of the neuromuscular junction mediated by B cells and associated with acetylcholine receptor (AChR) or muscle-specific receptor tyrosine kinase (MuSK) antibodies, which is characterized by muscle weakness and fatigability”

Author Response

1/ Atypical hemolytic uremic syndrome is still mentioned in the keywords

- Pompe disease is not mentioned in the keywords

We have deleted Atypical hemolytic uremic syndrome and added Pompe diseases in the keywords.

2/ Duchenne M.D.: The sentence "The calcium regulation may also be affected, with a pathological entrance of calcium into the muscle fibers. Excess can activate calpains and they promote muscle proteolysis." must be omitted. The matter is already said in the sentence above “This will in turn cause an increased calcium influx, with the activation of the calcium-dependent proteases and the subsequent digestion of the cytoskeletal and myofibrillar proteins.”

We have deleted the phrase that was repeated.

3/ DMD: Cite in the paragraph about deflazacort recent paper by Guglieri et al. Effect of Different Corticosteroid Dosing Regimens on Clinical Outcomes in Boys with Duchenne Muscular Dystrophy: A Randomized Clinical Trial. JAMA. 2022 Apr 19;327(15):1456-1468.

We have added the information regarding this study, and we have emphasized that following this clinical trial, the administration of corticotherapy had different efficacity also depending on the mode of administration (the patients receiving intermittent doses had worse outcome than those receiving continuous doses).

4/ hATTR: Therefore, the TTR mRNA undergoes the process of catalytic degradation, and the serum protein levels are reduced [42].

The phrase has been modified.

5/ Myasthenia gravis: The first sentence “Myasthenia gravis is an autoimmune chronic disease of the neuromuscular junction, mediated by anti-MUSK antibodies targeting the acetylcholine receptor (CchR), characterized by muscle weakness and fatigability.” Is wrong. 

Change in: “Myasthenia gravis is an autoimmune chronic disease of the neuromuscular junction mediated by B cells and associated with acetylcholine receptor (AChR) or muscle-specific receptor tyrosine kinase (MuSK) antibodies, which is characterized by muscle weakness and fatigability”

The false information has been revised and corrected in consequence.
